# Scorpion Peptide Smp24 Exhibits a Potent Antitumor Effect on Human Lung Cancer Cells by Damaging the Membrane and Cytoskeleton In Vivo and In Vitro

**DOI:** 10.3390/toxins14070438

**Published:** 2022-06-28

**Authors:** Ruiyin Guo, Junfang Liu, Jinwei Chai, Yahua Gao, Mohamed A. Abdel-Rahman, Xueqing Xu

**Affiliations:** 1Department of Pulmonary and Critical Care Medicine, Zhujiang Hospital, Southern Medical University, Guangzhou 510280, China; 21920154gry@i.smu.edu.cn (R.G.); ljf00157@163.com (J.L.); 2Guangdong Provincial Key Laboratory of New Drug Screening, School of Pharmaceutical Sciences, Southern Medical University, Guangzhou 510515, China; vivian20@i.smu.edu.cn (J.C.); gaoyahua7721@163.com (Y.G.); 3Zoology Department, Faculty of Science, Suez Canal University, Ismailia 41522, Egypt; mohamed_hassanain@science.suez.edu.eg

**Keywords:** lung cancer, antimicrobial peptide, scorpion, venom, tumor

## Abstract

Smp24, a cationic antimicrobial peptide identified from the venom gland of the Egyptian scorpion *Scorpio maurus palmatus*, shows variable cytotoxicity on various tumor (KG1a, CCRF-CEM and HepG2) and non-tumor (CD34^+^, HRECs, HACAT) cell lines. However, the effects of Smp24 and its mode of action on lung cancer cell lines remain unknown. Herein, the effect of Smp24 on the viability, membrane disruption, cytoskeleton, migration and invasion, and MMP-2/-9 and TIMP-1/-2 expression of human lung cancer cells have been evaluated. In addition, its in vivo antitumor role and acute toxicity were also assessed. In our study, Smp24 was found to suppress the growth of A549, H3122, PC-9, and H460 with IC_50_ values from about 4.06 to 7.07 µM and show low toxicity to normal cells (MRC-5) with 14.68 µM of IC_50_. Furthermore, Smp24 could induce necrosis of A549 cells via destroying the integrity of the cell membrane and mitochondrial and nuclear membranes. Additionally, Smp24 suppressed cell motility by damaging the cytoskeleton and altering MMP-2/-9 and TIMP-1/-2 expression. Finally, Smp24 showed effective anticancer protection in a A549 xenograft mice model and low acute toxicity. Overall, these findings indicate that Smp24 significantly exerts an antitumor effect due to its induction of membrane defects and cytoskeleton disruption. Accordingly, our findings will open an avenue for developing scorpion venom peptides into chemotherapeutic agents targeting lung cancer cells.

## 1. Introduction

Lung cancer, especially non-small cell lung cancer (NSCLC), has become the major cause of cancer death around the world, resulting in a mortality of more than 1.7 million annually [1]. Although chemotherapy, radiation, targeted therapy, and immune therapy have improved the survival rate of patients, lung adenocarcinoma is still one of the most aggressive and rapidly malignant types of cancer [1]. Therefore, developing new anticancer drugs with high selectivity, low toxicity, and adequate membrane permeability is of great importance.

Antimicrobial peptides (AMPs) are generally acknowledged as important components of the innate immune defense and are the most promising candidates to replace conventional antibiotics. Approximately 3000 native AMPs have been purified and identified from animals, plants, protozoa, bacteria, fungi and protists [2]. Of these peptides, 230 members among them are grouped into anticancer peptides (ACPs) that possess antitumor activity [2]. Most of ACPs target cell membranes rather than specific receptors, which make them less possible for cancer cells to develop drug resistance compared with traditional chemotherapy [3,4]. Therefore, the development of these peptides into potent anticancer agents, whether alone or in combination with other classical chemotherapy, has been considered as an innovative strategy for cancer treatment [3]. The action mechanism of most ACPs depends on the electrostatic interactions between cationic peptides and anionic lipids on cancer cell membranes [3,4]. As a result, they selectively disrupt the negatively charged plasma membranes of cancer cells without affecting zwitterionic cell membranes of non-cancer cells [3,4]. Furtherly, tumor cell migration is essential for invasion and dissemination from primary solid tumors [5]. Tumor migration suppression is also a therapy strategy and some AMPs, such as the short peptide RK1 from *Buthus occitanus*, can inhibit tumor cell migration and show therapeutic potential [6].

Scorpion venoms reveal potent suppressing effects on various types of cancers including lung, hepatoma, leukemia, breast, neuroblastoma, prostate, pancreas, glioma and lymphoma [7]. Our previous studies have described that Smp24, an amphipathic cationic AMP isolated from the Egyptian scorpion *Scorpio maurus palmatus*, can suppress microbes by membrane disruption [8,9]. The peptide shows stronger cytotoxicity against leukemic tumor cell lines (KG1-a and CCRF-CEM) than the normal cell lines (CD34^+^, HRECs and HACAT) [10], which indicates that Smp24 may be a potential ACP. However, its effects and underlying mechanism on lung cancer cells remain to be clarified. In this study, we first elucidated the cytotoxicity of Smp24 on human lung cancer cells via membrane destruction. Furthermore, its effects on the migration ability of A549 cells via the regulation of filamentous actin (F-actin) and the alteration of MMP-2/9 and TIMP-1/2 protein expressions were assessed. Finally, Smp24 showed antitumor efficacy and low acute toxicity in the A549 xenograft mice model. To the best of our knowledge, this is the first detailed study describing the potent in vitro and in vivo efficacy of Smp24 against lung cancer cells.

## 2. Results

### 2.1. Smp24 Inhibits the Proliferation of Human Lung Cancer Cells

MTT was used to measure the cytotoxic activity of Smp24 against four human lung cancer cell lines. The IC_50_ values and selectivity index of Smp24 for 24 h treatment against A549, H3122, PC-9 and H460 cells were about 4.06 ± 0.60, 4.64 ± 0.18, 6.34 ± 0.53, and 7.07 ± 0.81 µM, and 3.61, 3.16, 2.32, and 2.07, respectively (Figure 1A and Appendix A). Nonetheless, Smp24 exhibited less inhibitory potency toward the proliferation of normal cells as evidenced from the higher IC_50_ value of around 14.68 ± 0.79 μM for MRC-5 cells (Figure 1A), indicating that it is more selective toward cancer cells. Furthermore, the IC_50_ value of Smp24 for 24 h treatment against A549 is lower than 31.98 ± 1.79 µM of cisplatin (Figure 1A and Appendix A). Because the A549 cell line is most sensitive to Smp24 and NSCLC accounts for about 80–85% of human lung carcinoma [1], the NSCLC cell line A549 was selected for further investigation of the antitumor effect of Smp24. EdU incorporation assay revealed that Smp24 dramatically decreased the EdU fluorescence intensities of A549 cells compared with untreated cells, which further suggested that Smp24 inhibited cell proliferation in a dose-dependent manner (Figure 1B). The morphology of A549 cells incubated with Smp24 for 24 h is shown in Figure 1C. The control cells formed normal and fusiform shape with smooth surfaces. However, A549 cells treated with Smp24 were obviously smaller and rounder than the control cells. Moreover, their number decreased in a concentration-dependent manner with many floated cells and cellular debris. The above data suggest that Smp24 has both cytotoxic and cytostatic effects on lung cancer cells.

### 2.2. Smp24 Induces Necrosis of A549 Cells by Membrane Damage

To decipher the mechanism behind the antiproliferative effects of Smp24, its necrotizing effect was examined. The release of LDH was increased in Smp24-treated cells in both dose- and time-dependent manners (Figure 2A). The release of LDH evidenced cell membrane leakage throughout. Thus, SEM technology was used to observe morphological alteration of the cell membranes induced by Smp24. As shown in Figure 2B, the control cells displayed complete and smooth cell membranes, whereas Smp24-treated cells appeared with fragmented cell membranes and curly edges. The above findings indicated that the decrease in cell viability induced by Smp24 resulted from its cytolytic effect. To further confirm the membrane permeability changes following treatment with Smp24, calcein AM release assay was carried out. As shown in Figure 2C, Smp24 exposure concentration dependently accelerated the release of calcein AM from A549 cells, indicating the damage of membrane integrity in A549 cells. CoCl_2_ is a quencher of cytosolic calcein fluorescence to selectively label mitochondria. Consistently, in the presence of CoCl_2_, the cytosolic fluorescence was reduced from 90.70% to 74.70%, while for mitochondria, it increased from 1.5% to 8.17% when compared to calcein AM alone. Furthermore, this change in trend of the fluorescence intensity in the cytoplasm and mitochondria was further strengthened by Smp24 in a concentration-dependent manner when compared with the control without Smp24 treatment (Figure 2C). Additionally, it is the necroptosis inhibitor necrostatin-1 rather than mitochondrial permeability transition pore inhibitor, Cyclosporine A, that can inhibit Smp24-induced proliferation and necrosis of A549 (Figure 2D,E). To further confirm the membranolytic effect of Smp24 on the nuclear membrane, DAPI staining assay was performed. As shown in Figure 2F, the untreated A549 cells were found to contain intact nuclei. However, the volume of A549 cells treated by Smp24 for 24 h was reduced, and the nuclear integrity was broken. Furthermore, in FITC channel observation, all cells showed a high green fluorescence due to the peptide in the cells and some green fluorescence co-localized blue fluorescence-derived DAPI staining, indicating that a portion of Smp24 entered into the nucleus, thereby damaging the cell nucleus (Figure 2F). Together, these results suggested that Smp24 induced non-specifically cellular and mitochondrial membrane damage in A549.

### 2.3. Smp24 Inhibits Motility by Damaging the Cytoskeleton and Altering MMP-2/-9 and TIMP-1/-2 Expression

Structural stability of cells and the integrity of their plasma membrane mostly rely on the dynamics and function of the cytoskeleton, which allow cells to maintain or adaptably modify their morphology to facilitate cell division, motility, and other biological activities [11,12]. Thus, the cytoskeleton was examined via staining F-actin with rhodamine–phalloidin. As displayed in Figure 3A, the cells without Smp24 treatment were smooth with obvious staining of flat microfilament bundles that were collateral to the edge of the cells. However, F-actin microfilaments became disorganized and polarized feathers of cytoskeleton variation following treatment with Smp24. Especially, after co-culture with 5 μM Smp24 for 24 h, the microfilament bundles of A549 cells were shortened and disordered with visibly red fluorescence. The above data indicate that Smp24 suppresses the cell motility via the regulation of F-actin type in A549 cells.

Previous reports have confirmed that the cytoskeleton is crucially involved with cell motility, which is importantly responsible for high cancer mortality, and its crucial steps include cell migration and invasion [13,14]. For this reason, the wound-healing and transwell assays were performed to measure the effects of Smp24 on migration and invasion of A549 cells. In comparison with the control cells, A549 cell migration was suppressed by Smp24 in a dose-dependent manner after treatment for 24 h, and the suppression rates of 0.3, 0.6 and 1.2 μM of Smp24 were about 62.23%, 78.63%, and 94.52%, respectively (Figure 3B,C). Consistently, Smp24 decreased the invasion of A549 cells at a concentration-dependent manner in the invasion assay, respectively (Figure 3D). Therefore, Smp24 could effectively inhibit cell invasion and migration of A549 cells. The regulation proteins of the extracellular matrix, such as MMPs and TIMPs, play important roles in cell motility [13]. Therefore, the influence of Smp24 on the mRNA expression of MMP-2/-9 and TIMP-1/-2 were measured by qPCR. As shown in Figure 3E, the expressions of TIMP-1/-2 mRNA were significantly upregulated while the expressions of MMP-2/-9 mRNA were significantly downregulated in a concentration-dependent manner following incubation with Smp24 for 12 h. In short, these results confirmed that Smp24 is in a position to suppress A549 cell invasion and migration.

### 2.4. Smp24 Inhibits Tumor Growth in Mice

The in vivo pharmacodynamic effects of Smp24 were assessed using A549 xenograft mice model. As indicated in Figure 4, in comparison to the control group, the tumor volume and weight in the Smp24-treated group were reduced by approximately 65.4% and 64.3%, respectively (Figure 4B–D). However, the treatment with Smp24 did not affect the weight changes of body and important organs such as heart, lung, liver, spleen and kidney (Figure 4E,F). As shown in Figure 4G, HE staining displayed that tumor cells in the control group generally were enlarged with fairly complete structure and regular shapes, while massive cell shrinkage and the inflammatory cell infiltration appeared in the Smp24-treated group. Furthermore, immuno-histochemical analysis demonstrated that the expression of cleaved caspase-3 (brown area) in the Smp24-treated group was dramatically enhanced when compared to the control group (Figure 4H). The above results suggested that Smp24 can exert potent antitumor effects in vivo.

### 2.5. Smp24 Has No Hepatotoxicity and Nephrotoxicity in Acute Toxicity Doses

In the acute toxicity test of mice, physiological saline was selected as the normal control group. Doses of 5 and 10 mg/kg were chosen for the acute toxicity test. After the first day’s injection, there was no animal death in any group. Plasma AST, ALT, BUN and Cre are key biomarkers reflecting liver and kidney damage in the clinic [15]. All of them did not change significantly in the mice of the 5 and 10 mg/kg Smp24 groups after injection 48 h (Figure 5A–D). Compared with the important organs weights of mice pre-injection, the heart, lung, liver, spleen and kidney weights of mice in the 5 and 10 mg/kg Smp24 groups did not change significantly (Figure 5E–I). Therefore, the results indicated that Smp24 might not have hepatotoxicity and nephrotoxicity in the acute toxicity doses.

## 3. Discussion

Despite the great improvement that has been achieved in its treatment in recent decades, lung cancer remains the most lethal type of cancer and still is one of the biggest threats to the health care system globally [1]. AMPs derived from natural or designed peptides possess selective cytotoxicity against various tumor cells [16]. Smp24 is an amphipathic and cationic AMP from Egyptian scorpion *Scorpio*
*maurus palmatus*, which exhibits the cytotoxicity to leukemic tumor cell lines [10]. In line with the previous study, we have found that Smp24 preferentially inhibits the proliferation of A549, H3122, PC-9 and H460 cells in vitro (Figure 1A). Moreover, its antitumor capability is further proven in the xenograft mice (Figure 4).

A previous study has also proven that Smp24 can induce pore formation of membrane [9]. In line with them, our compelling evidence from LDH release, calcein AM staining and SEM observation (Figure 2) demonstrate that cationic Smp24 as a pore-forming peptide can interact with tumor cell membranes, elevate their permeability, and induce pore formation in the plasma membrane and cell death [3,17,18]. Recent findings suggest that some cationic AMP such as Brevivin-1RL1 [19], KL15 [20], D-K6L9 [21] and P18 [22] can exhibit their induction of tumor necrosis via membrane destruction. Furthermore, after exposure to A549 cells for 24 h, Smp24 causes mitochondrial damage (Figure 2C) and cytoskeleton disruption (Figure 3A). In addition, Smp24 can distribute into the cell nucleus, thereby damaging the nucleus (Figure 2F), and consequently resulting in cell death. Finally, the signal pathway inhibitors such as Necrostatin-1 can inhibit Smp24-induced proliferation and necrosis of A549 (Figure 2D,E). Cyclosporine A, the permeability transition inhibitor, cannot inhibit the cytotoxicity of Smp24 to A549 cells (Figure 2D,E), suggesting that the non-specific mitochondrial membrane disruption of Smp24 should be responsible for its induction of the mitochondrial dysfunction. These data further suggest that the cytotoxicity of Smp24 to tumor cells might result from its damage of the plasma membranes and other intracellular cascades.

The different charge distributions of the cell membrane surface between cancer and non-cancer cells may partially contribute to the selectivity of AMPs. For example, the rich anionic components such as heparan sulfate, PS, and sialylated gangliosides provide a net negative charge on the surface of cancer cells. For this reason, AMPs can electrostatically bind cancer cell membranes and especially kill tumor cells [3]. Consistently, a series of short amphiphilic triblock AMPs shows that anti-tumor effects and K_4_F_6_K_4_ with high charge most strongly affects A549 cells among them. Differently, although Smp24 has significantly lower overall charge (+3) than K_4_F_6_K_4_, the IC_50_ value (32.21 μM) of K_4_F_6_K_4_ against A549 cells is higher than 4.06 μM of Smp24. It is reported that the net charge need not be as high as possible to improve the anticancer activity, and some mutant peptides with low charge showed strong activity among a series of peptides [23]. Thus, net charge and anticancer activity are not always positively correlated. Instead, some α-helical peptides with cationic and amphipathic character derived from animal venoms can penetrate the membrane of tumor cells despite their difference in spatial structure and the content of disulfide bonds. Thus, other physical and chemical properties such as hydrophobicity and helicity may also contribute to the antitumor capability of AMPs [24,25].

Generally, the cytoskeleton cannot counteract the external pressure from the extracellular environment, the plasma membrane may be irreversibly deformed, leading to plasma membrane leakiness [12]. Specifically, actin fragmentation eliminates the cytoskeletal support of the plasma membrane, which then causes the appearance of pores in the plasma membrane [11]. Plasma membrane pores unbalance the controlled trafficking of ions (e.g., calcium ion), metabolites and waste products in and out of the cell and, thereby, disrupt the intracellular homeostatic conditions. Clearly, calcium concentrations that are above critical are cytotoxic, as they lead to mitochondrial failure [26]. In this study, both the cell membrane and mitochondrial membrane were damaged after Smp24 treatment (Figure 2). Hence, persistent perturbations to the cytoskeleton may lead to breakdown of cell function, permeabilization of the plasma membrane, loss of cell homeostasis, and eventual cell death [12]. 

Cell migration is a highly dynamic process controlled by the cytoskeleton, which mainly comprises the actin microfilaments, microtubules and intermediate filaments. During migration, cells polarize and form protrusions at the front where new adhesions are formed, which transmit the traction forces required for movement. All of these steps are coupled to major cytoskeletal rearrangements and are controlled by a wide array of signaling cascades [27]. In addition, F-actin is a functional indicator of cell migration and can mostly affect the motility of cancer cells [28,29]. In our experiment, after treatment with Smp24 for 24 h, both F-actin reorganization and cell motility are inhibited (Figure 3A–D). Recent studies have linked the mitochondrial ROS increase to suppression of lung cancer cell motility [30,31,32]. Moreover, mitochondria dysfunction, ROS accumulation and perturbation of F-actin fibers in the liver cancer cells treated by antitumor molecules coincidently occurred [33,34]. Considering that Smp24 can increase the contents of ROS in A549 cells (data not shown), Smp24 damages the mitochondrial membrane and increases ROS content, which disrupts the cytoskeleton. Focal adhesions (FAs) are dynamic signaling structures that anchor the cytoskeleton to the extracellular matrix via numerous effector proteins such as integrins, cadherins and other adhesion molecules [35]. Accordingly, microtubules can also adjust the integrin activation, the recruitment of specific adhesion complex components, and the assembly of FAs [36,37]. Furtherly, the assembly and disassembly of Fas are chiefly regulated by focal adhesion kinase (FAK), which can be inhibited by cytoskeleton disorganization. In addition, FAs disassembly has been reported to require dephosphorylation of P-FAK [38], and the stiffness-dependent activations of the FAK–p130Cas–Rac and PI3K-Akt pathway lead to the changed expression and distribution of N-cadherin and integrins in cells [39,40]. Considering that Smp24 can inhibit the phosphorylation of FAK and PI3K (data not shown) and damage the cytoskeleton, we can deduce the reduced adhesive potential of these cells and inhibit the mobility of cells, which is consistent with the effects of polypeptides reported by Xia et al. [41]. The migration and invasion of NSCLC, which are greatly affected by the MMPs protein family, especially MMP-2 and MMP-9, are important for its metastasis to other body regions such as bone, brain and spleen. Thus, their suppression is a matter of importance for the treatment of NSCLC [1,13]. Smp24 is demonstrated to own the obvious suppression effects on the migration and invasion of A549 cells (Figure 3B–D). Meanwhile, Smp24 can dramatically decrease the mRNA levels of MMP-2/-9 while increasing those of TIMP-1/-2 in A549 cells (Figure 3E). These data confirm that Smp24 can markedly restrain the metastasis of A549 cells. Therefore, the motility suppression of lung cancer cells by Smp24 may be associated with its changes of mRNA levels of MMP-2/-9 and TIMP-1/-2 plus F-actin reorganization.

For most peptides, despite the cogent evidence about their antitumor effects in vitro, their clinical applications have met with major obstacles owing to several issues, including their non-specific cellular cytotoxicity and rapid degradation in the blood [42]. In order to further verify the antitumor effects of Smp24 in vivo, a tumor-burdened experiment was carried out. In our study, 18 d after treatment of 2 mg/kg Smp24, the tumor weight and volume were reduced about 64.3% and 65.4%, respectively, which demonstrate its stability and antitumor potency in vivo (Figure 4B–D). Furthermore, it does not affect the weight changes of body and important organs such as heart, liver, spleen, lung and kidney (Figure 4E,F), confirming its low cytotoxicity to normal mammalian cells and biological safety in vivo. Melittin is a cationic AMP with 26 amino acids and presents antitumor effects toward various tumor cells via occurrence of necrosis and motility [43]. However, subcutaneous injection of melittin at doses of 0.5 and 10 mg/kg significantly suppresses NSCLC tumor growth by 27% and 61%, respectively [44]. Considering that Smp24 contains 24 amino acids in its primary sequence and has stronger antitumor capacity against NSCLC than melittin in vivo, and the high dose of Smp24 (10 mg/kg) lacks damaging effects on the liver and kidney (Figure 5), Smp24 has lower synthesis cost and is more efficient than melittin. Together, Smp24 has the anti-A549 cell effects in vivo without remarkable toxicity, which makes it more in reference to clinical trials. However, most natural peptides are difficult to directly apply in clinics without modification to improve their therapeutic efficacy and stability in plasma. The effects of Smp24 with cyclization, hybridization and nanodrug modification on tumor cells should be explored in the future.

## 4. Conclusions

In conclusion, the present findings indicate that Smp24 exerts a cytotoxic effect in vitro and an antitumor activity in in vivo models of lung carcinoma cells by suppression of cell motility and induction of necrosis. Mechanistically, Smp24 can destroy the integrity of the A549 cell membrane, mitochondrial and nuclear membrane, inhibit cytoskeleton organization and change the expression of MMP-2/-9 and TIMP-1/-2 in A549 cells. This discovery will extend the antitumor mechanism of AMPs and open an avenue for developing scorpion venom peptides into chemo-therapeutic agents targeting lung cancer cells.

## 5. Materials and Methods

### 5.1. Animals and Ethics Statement

A total of 10 BALB/c nude mice of both sexes (18–20 g, 6 weeks) were acquired from the Laboratory Animal Center of Southern Medical University and were raised in a specific pathogen-free mini-barrier system of the central animal facility of Southern Medical University under controlled conditions (60% humidity, 21 ± 2 °C room temperature, and 12 h light-dark cycle). The experimental protocols (ethical approval number: L2019226 on 11 November 2019) involving animals were approved by the Animal Ethics Committee of Southern Medical University and were implemented in the light of the international regulations for animal research. The animals were placed in an induction chamber, and anesthesia was induced with 5% isoflurane before sacrificed by cervical dislocation.

### 5.2. Chemicals and Cell Culture

RPMI-1640, fetal bovine serum (FBS), phosphate-buffered saline (PBS) and trypsin were all obtained from Gibco (New York, NY, USA). A549, H3122, PC-9, H460 and MRC-5 cells were purchased from the American Type Culture Collection (Manassas, WV, USA). Cells were grown in RPMI-1640 medium containing 1% penicillin–streptomycin and 10% FBS under the condition of 37 °C and 5% CO_2_. Lactate dehydrogenase (LDH) release assay kit, rhodamine–phalloidin and DAPI were obtained from Beyotime Institute of Biotechnology, Shanghai, China. Smp24 (IWSFLIKAATKLLPSLFGGGKKDS) was synthesized by GL Biochem Ltd. (Shanghai, China) and purified with an Inertsil ODS-SP (C-18) RP-HPLC column (Shimazu, Japan) to over 95% purity. The high-purity peptide was collected, lyophilized, and further identified by MALDI-TOF mass spectrometry (Appendix A). The theoretical mass of 2578.09 Da of the peptide coincides with the experimentally determined one, 2578.30 Da.

### 5.3. Cell Viability and Proliferation Assays

MTT assay was performed to analyze cellular viability as previously reported by us [45]. Briefly, A549, H3122, PC-9, H460 and MRC-5 cells (1 × 10^4^ cells/well) were grown in 96-well plates and exposed to the different concentrations of Smp24 (1.25–20 μM) with 24 h. To determine which signal pathway is primarily responsible for the cytotoxic effects of Smp24 on A549 cells, various inhibitors including 40 μM Necrostatin-1, 40 μM Z-DEVD-FMK, and 1 μM cyclosporine A were pre-incubated with A549 cells for 30 min before 5 μM Smp24 was incubated with the cells for 12 h. After incubation, 10 µL MTT (5 mg/mL) was added before incubation for 4 h at 37 °C in the dark. Subsequently, the cell medium was discarded, and 200 µL DMSO was applied into each well before the absorbance value at 490 nm was determined via the microplate reader (Tecan Company, Männedorf, Switzerland). Cell proliferation was further detected by the BeyoClick™ EdU cell proliferation kit with Alexa Fluor 488 (Beyotime Institute of Biotechnology, Shanghai, China) according to the protocol of manufacturer. In brief, A549 cells (2 × 10^5^ cells/well) were grown in 6-well plates and exposed to different concentrations of Smp24 (0, 1.25, 2.5 and 5 μM) for 24 h. Then, the cells were incubated with EdU for 2 h and stained with Alexa Fluor 488 for 30 min in the dark before cell fluorescence intensity was measured by flow cytometry (Becton Dickinson Company, Franklin Lakes, NJ, USA). The experiments had been performed in triplicate.

### 5.4. Membrane Integrity Assay

To examine the membrane integrity, A549 cells were stained with calcein AM following the manufacturer’s manual (Beyotime Institute of Biotechnology, Shanghai, China). Briefly, 1 × 10^5^ A549 cells were cultured in a 12-well plate overnight. Then, the cells were exposed to Smp24 (0, 1.25, 2.5, and 5 μM) for 24 h and detached by trypsinization, washed with PBS, followed by incubated with 1 µM calcein AM at 37 °C for 30 min. After calcein, AM in the medium was cleared and cells were washed with PBS, the levels of calcein AM were detected by flow cytometry (Becton Dickinson Company, Franklin Lakes, NJ, USA). To measure the integrity of the mitochondrial membrane, after cells were incubated with calcein AM for 30 min and washed with PBS for three times, 1 μM CoCl_2_ was added into the cells and incubated at 37 °C for another 15 min, followed by flow cytometry analysis. All experiments had been detected in triplicate.

### 5.5. Cell Morphology Observation

First, 2 × 10^5^ A549 cells were seeded onto a 6-well plate overnight. Subsequently, the cells were exposed to Smp24 at the different concentrations (0–10 μM) for 24 h before the cellular morphology was examined under the inverted phase contrast microscope (100× magnification). Approximately 3 single pictures of each well were captured.

### 5.6. LDH Release Assay

The LDH release assay was carried out with a commercial kit (Beyotime Institute of Biotechnology, Shanghai, China) following the manufacturer’s manual. In short, A549 cells were incubated with Smp24 (1.25, 2.5, 5, 10 and 20 μM) for 12, 24, and 48 h, respectively. Then, 10 µL of LDH release solution was applied per well and incubated for 1 h. The supernatant was transferred to a new plate and mixed with 60 µL of substrate solution per well by gentle shaking for 30 min in the dark. In some experiments, A549 cells were pre-incubated with the inhibitors (40 μM Necrostatin-1, 40 μM Z-DEVD-FMK, and 1 μM cyclosporine A) for 30 min before the cells were incubated with 5 μM of Smp24 for 12 h. Thereafter, a microplate reader (Tecan Company, Männedorf, Switzerland) was used to measure the absorbance value at 490 nm. All experiments were conducted in triplicate.

### 5.7. Scanning Electron Microscopy Analysis

A549 cells (1.2 × 10^5^ cells/well) were cultured on the glass coverslips inserted in a 12-well plate for 24 h. Next, a range of concentrations of Smp24 (0, 2.5, and 5 μM) was added to the 12-well plate and co-cultured for 24 h. The cells without Smp24 were considered as the negative control. The cells were sequentially fixed with 4% glutaric dialdehyde at room temperature for 2 h and 2.5% glutaric dialdehyde at 4 °C for 8 h, followed by dehydrating with a series of gradient ethanol/water solutions. Subsequently, the samples were dried and coated with gold before observation with Phenom ProX instrument at 15 kV. The experiments were conducted in triplicate.

### 5.8. Fluorescence Microscopy Analysis

For F-actin reorganization analysis, 7 × 10^4^ A549 cells were seeded in a 24-well plate overnight and incubated with Smp24 (0, 2.5, and 5 μM) for 24 h. After being fixed with 4% PFA, the cell F-actin was stained with rhodamine–phalloidin for 30 min, washed with PBS, and then stained with DAPI for another 10 min. The F-actin was observed under fluorescence microscope at 400× magnification after washing.

To further locate Smp24 within cells, after treatment with 5 μM FITC-labeled Smp24 at 37 °C for 24 h, the cells were washed with PBS, fixed with 4% paraformaldehyde (PFA) for 30 min, stained with DAPI for 10 min and followed by observation with fluorescence microscope at 400× magnification. About 3 single-plane pictures for each well were obtained.

### 5.9. Cell Motility Assay

Cellular motility was determined using both the scratch migration and transwell invasion assays as previously described by us [45]. For the scratch migration assay, 2 × 10^5^ A549 cells were plated onto 6-well plates overnight. After scratching the cell monolayer with a P10 pipette tip on the second day, the cell fragments were removed by washing with PBS. A549 cells were incubated in serum-free 1640 medium with Smp24 (0, 0.3, 0.6 and 1.2 μM) for 24 h. Three single-plane photos were used to detect scratch widths at 0 h (W_0_) and 24 h (W_t_). Per point, the migration index (M_I_) was calculated as M_I_ = W_t_/W_0_. The cell transwell invasion assay was conducted with a transwell plate (8 µm pore size; Corning, Inc., New York, NY, USA). The upper compartment was seeded with 5 × 10^4^ cells in 100 µL 1640 serum-free medium with Smp24 (0, 0.3, 0.6, and 1.2 μM). The lower compartment was added by 500 μL of 1640 containing 10% FBS as an attractant for 24 h. Subsequently, the noninvasive cells (the upper side) were removed with cotton swabs, and the invasive cells on the lower side were fixed with 4% formaldehyde, followed by crystal violet staining before being photographed under a light microscope (200× magnification). Next, the crystal violet was dissolved by 100 µL of 10% (*v*/*v*) acetic acid, and the absorbance value at 570 nm was measured to quantify cell invasion via a microplate reader (Tecan Company, Männedorf, Switzerland). All experiments were conducted in triplicate.

### 5.10. Quantitative Real-Time Polymerase Chain Reaction (qRT-PCR) Assay

First, 2 × 10^5^ A549 cells were grown in 6-well plates overnight and mixed with Smp24 (0, 0.3, 0.6, and 1.2 μM) for 12 h. Next, the cells were centrifuged and the precipitate was harvested for qRT-PCR to measure the mRNA levels of MMP-2, MMP-9, TIMP-1 and TIMP-2 with the qPCR instrument (Light Cycler480II, ROCHE Ltd., Basel, Switzerland) as reported previously by us (Table 1) [45]. To upregulate the expression of MMP-9, the cells were pretreated with PMA (50 nM) for 1 h before harvest. The reaction cycles for all genes were: 95 °C for 3 min, 40 cycles at 94 °C for 15 s and 60 °C for 45 s. *GAPDH* gene was quantified as a control to verify equal initial quantities of RNA and as an internal standard to quantify PCR products. Results were calculated with the 2^−ΔΔCT^ method, and target gene expression was standardized to *GAPDH*. All experiments were conducted in triplicate.

### 5.11. In Vivo Antitumor Experiments

First, 5 × 10^6^ A549 cells with viability over 90% were inoculated subcutaneously into the right flank per mouse. Tumor growth was examined daily by palpation and the diameter was measured with calipers in two planes. Tumor volumes were counted with the following formula: volume (mm^3^) = (smallest diameter)^2^ × (largest diameter)/2. Then, the mice were randomized into two groups (n = 5 mice/group) for physiological saline and Smp24 treatment. When the tumor grew to about 50 mm^3^, Smp24 (2 mg/kg) or physiological saline was administrated near the tumor site every three days for 18 days. The body weight of mice and tumor volumes were recorded daily. On the 18th day of administration, the tumors and organs (heart, liver, spleen, lung, kidney) were removed, weighed and photographed. After embedding in paraffin, the tumors were stained with HE staining reagent to analyze the morphological alterations with lung cancer cells. To further detect the tumor tissue apoptosis levels, the apoptosis-related cleaved caspase-3 was also examined by immunohistochemical staining followed by observation under inverted-phase contrast microscope (200× magnification). Optical density analysis was performed on the immunohistochemical images using Image J software. The results were acquired from five mice for each group.

### 5.12. Acute Toxicity Analysis

To detect the acute toxicity of Smp24 in vivo, the mice were weighed and randomly divided into three groups with 3 animals each, and Smp24 (5 mg/kg and 10 mg/kg) and saline were intraperitoneally administered into mice of different groups. At 48 h after injection, serum and organ tissues from mice were collected for blood biochemical detection and organ weight measurement, respectively.

### 5.13. Statistical Analysis

Statistical analysis was performed by one-way ANOVA with Bonferroni’s multiple comparison with GraphPad Prism 5.0 (GraphPad Software, Inc., La Jolla, CA, USA). Data were recorded as mean ± SEM. Statistical significance was shown as * *p* < 0.05, ** *p* < 0.01 and *** *p* < 0.001. 

## 6. Patents

The study has been patented, grant number 202110599012.0.

## Figures and Tables

**Figure 1 toxins-14-00438-f001:**
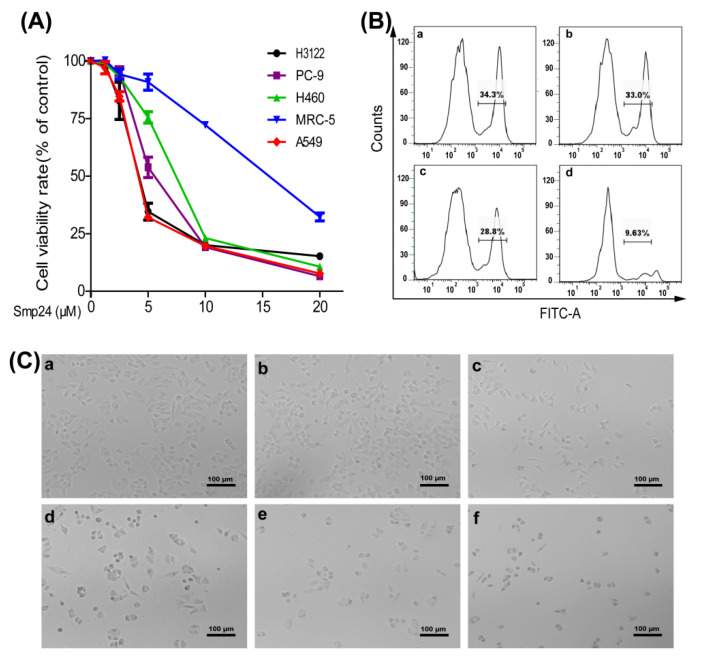
Viability and morphology changes of cancer cells induced by Smp24. (**A**) Viability of A549, H3122, PC-9, H460 and MRC-5 cells treated with Smp24 for 24 h. (**B**) Proliferation of A549 cells treated with the indicated concentrations of Smp24 for 24 h. Panels (**a**–**d**) are sequentially the cells treated with 0, 1.25, 2.5, and 5 μM of Smp24. (**C**) Cell morphology changes after treatment with Smp24 for 24 h. Panels (**a**–**f**) are sequentially the cells treated with 0, 1.25, 2.5, 5, 10 and 20 μM of Smp24. Scale bar, 100 μm. Data are normalized to control and presented as mean ± SEM (n = 3).

**Figure 2 toxins-14-00438-f002:**
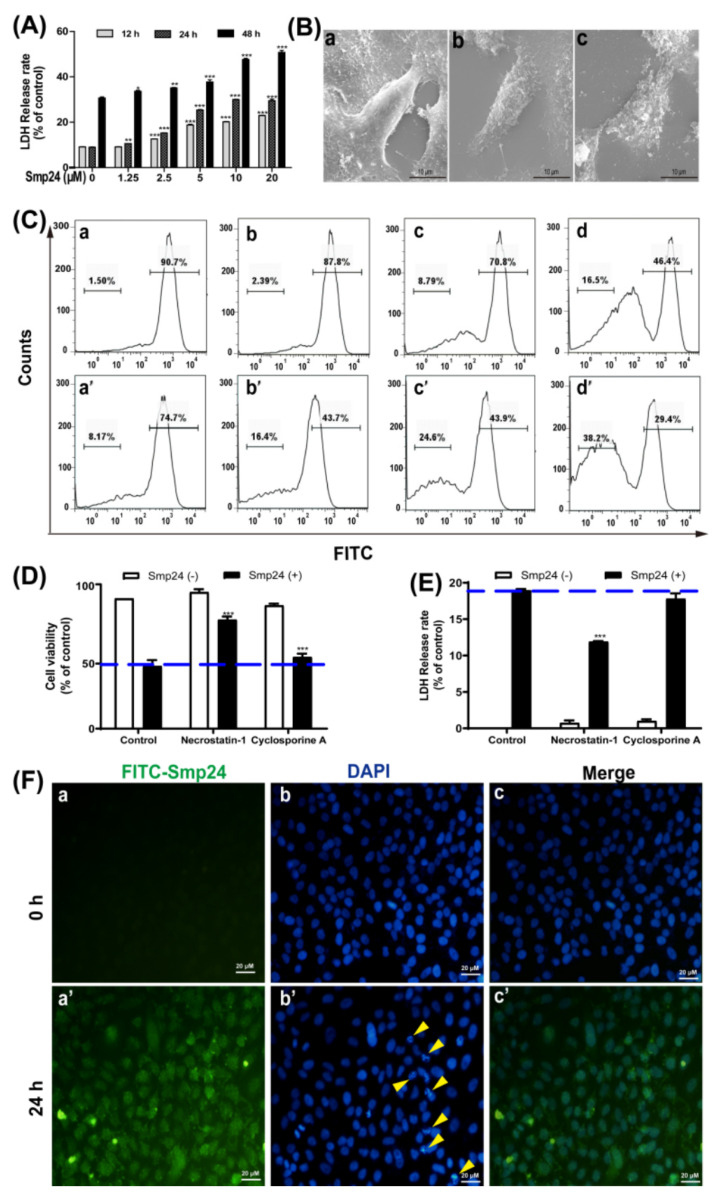
Necrosis induced by Smp24 in A549 cells. (**A**) LDH release of A549 cells induced by Smp24 (0–20 µM) for 12, 24 and 48 h. (**B**) SEM analysis of morphological structure of A549 cells. Panel (**a**): control A549 cells; panels (**b**,**c**): A549 cells treated by 2.5 and 5 µM of Smp24. Scale bar, 10 μm. (**C**) Representative flow cytometry analysis of calcein AM changes in A549 cells treated with Smp24 for 24 h. Panels (**a**–**d**): cells stained with calcein AM; panels (**a’**–**d’**): cells treated by calcein AM + CoCl_2_; Panels (**a**–**a’**,**b**–**b’**,**c**–**c’**,**d**–**d’**): cells treated by 0, 1.25, 2.5 and 5 µM of Smp24, respectively. Results are presented as mean ± SEM (n = 3). * *p* < 0.05, ** *p* < 0.01, *** *p* < 0.001 are considered statistically significant as compared to the control. (**D**,**E**) Effects of inhibitors on the viability and the LDH release of Smp24-treated A549 cells. A549 cells were pre-incubated with the inhibitors (40 μM Necrostatin-1, 1 μM Cyclosporine A) for 30 min before being further incubated with 5 μM Smp24 for 12 h. (**F**) Fluorescence microscope observation of 5 μM FITC-labeled Smp24 internalized in A549 cells after co-incubation for 24 h. (**a**,**b**) Control (no treatment) group; (**a’**,**b’**) 24 h treatment group; panels (**c**,**c’**) are the merged figure. Yellow arrow: nuclear fragmentation or the apoptotic body. Scale bar, 20 μm.

**Figure 3 toxins-14-00438-f003:**
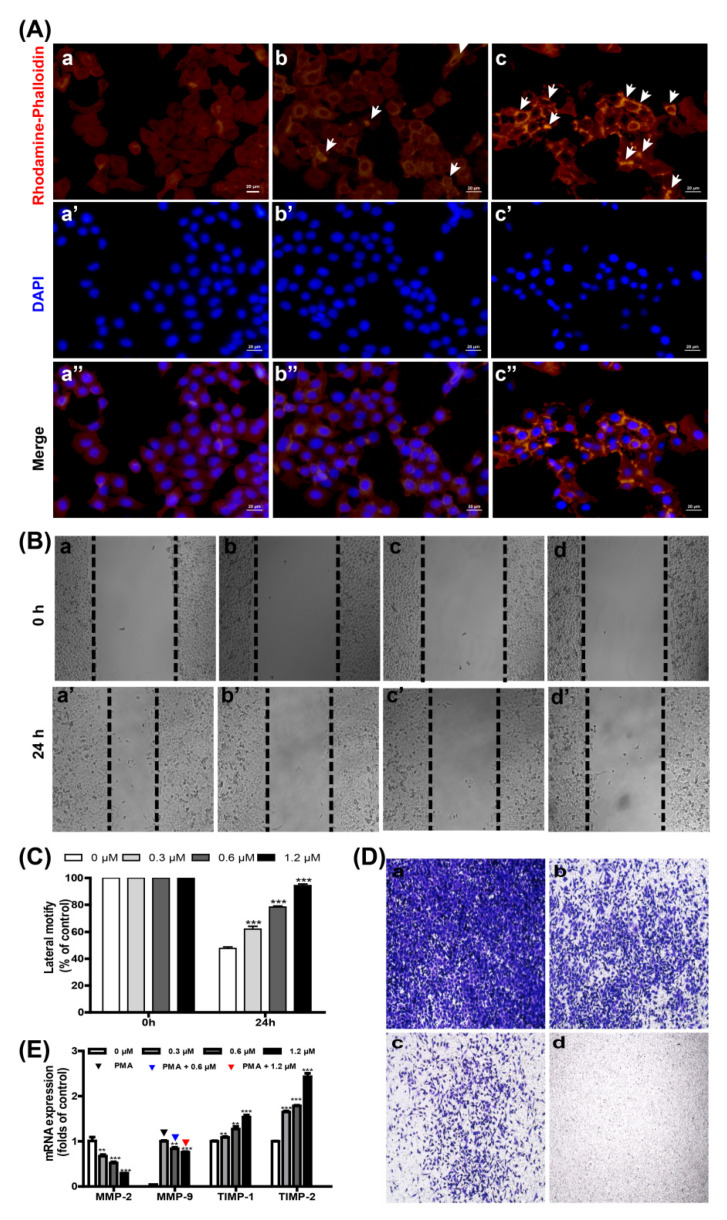
Effects of Smp24 on the motility and cytoskeleton reorganization of A549 cells. (**A**) Fluorescence staining images of F-actin. **Upper** panel: cells stained with rhodamine-phalloidin; **middle** panel: cells stained with DAPI; **lower** panel: the merging images of cells stained by rhodamine–phalloidin and DAPI. A549 cells were treated with Smp24 (0, 2.5, and 5 μM) for 24 h and successively stained by rhodamine-phalloidin as well as DAPI before fluorescence microscopy observation. White arrow: disordered microfilament bundles. Scale bar, 20 μm. (**B**) Representative pictures of A549 cells in the scratch migration assay at 0 and 24 h following incubation with Smp24 and PBS. Panels (**a**–**a’**,**b**–**b’**,**c**–**c’**,**d**–**d’**): the cells treated by 0, 0.3, 0.6 and 1.2 µM of Smp24, respectively. (**C**) Statistical analysis for the scratch migration assay. (**D**) Typical images of A549 cells in the transwell invasion assay following culture with Smp24 and PBS for 24 h. Panels (**a**–**d**): the cells treated by 0, 0.3, 0.6 and 1.2 µM of Smp24, respectively. (**E**) Statistical analysis of relative mRNA contents of MMP-2, MMP-9, TIMP-1 and TIMP-2. Results are mean ± SEM (n = 3). ** *p* < 0.01 and *** *p* < 0.001 are considered statistically significant as compared to the control group without Smp24.

**Figure 4 toxins-14-00438-f004:**
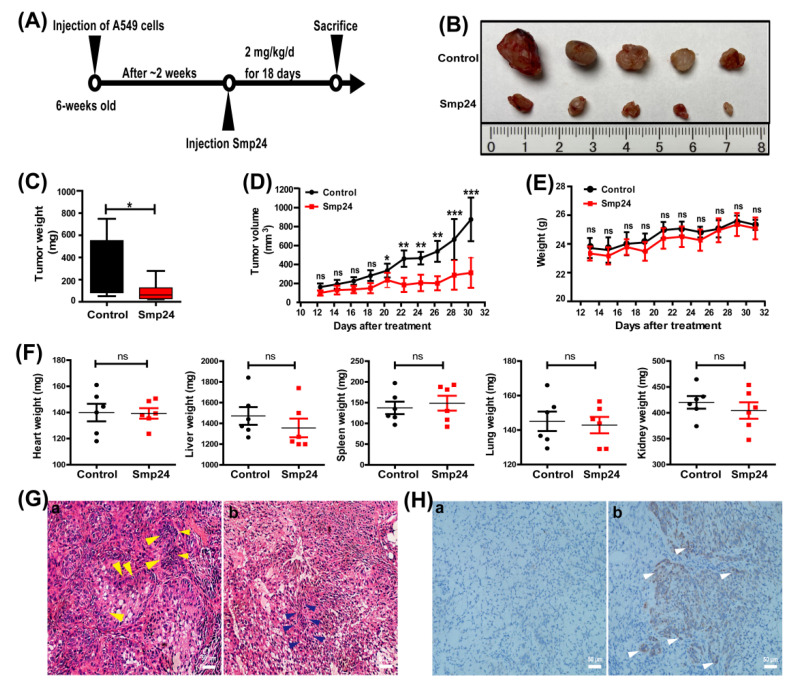
In vivo antitumor effects of Smp24. (**A**) Experimental schedule for xenograft mice. (**B**) Images of tumors from sacrificed nude mice. (**C**) Tumor weight. (**D**) Tumor volume. (**E**) Body weight change. (**F**) The weight of heart, liver, spleen, lung and kidney. (**G**) HE staining analysis of tumor tissue. Scale bar, 50 μm. (**H**) Immunohistochemical analysis of cleaved caspase-3 in tumor tissue from lung carcinoma xenografts. Panel (**a**): control group; panel (**b**): Smp24-treated group. Scale bar, 50 μm. Data presented are mean ± SEM (n = 5). ns: no significance, * *p* < 0.05, ** *p* < 0.01 and *** *p* < 0.001 are considered statistically significant as compared to the control group without Smp24 treatment. Yellow arrow: enlarged tumor cells. Blue arrow: shrinking cells. White arrow: positive apoptotic staining (brown areas).

**Figure 5 toxins-14-00438-f005:**
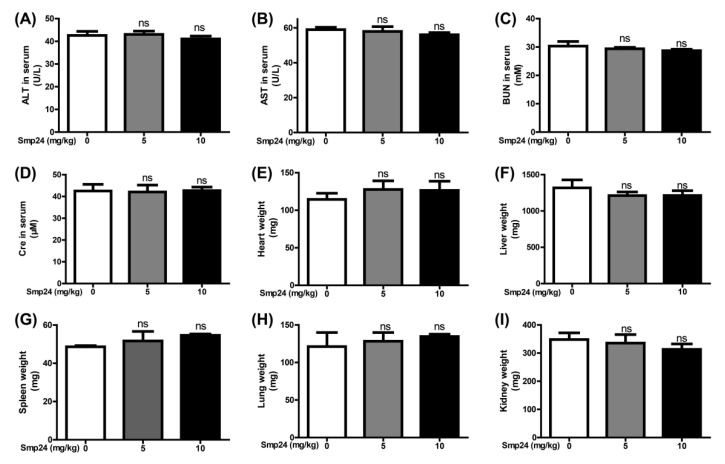
Acute toxicity analysis of Smp24. (**A**–**D**) The level of ALT, AST, BUN, Cre in serum of normal mice after 48 h treatment with 0, 5 and 10 mg/kg Smp24. (**E**–**I**) The weight of heart, liver, spleen, lung and kidney of mice after 48 h treatment with 0, 5 and 10 mg/kg Smp24. ns: no significance.

**Table 1 toxins-14-00438-t001:** The primers sequences of qRT-PCR.

Genes	Forward	Reverse
*GAPDH*	5′-CGGAGTCAACGGATTTGGTCGTAT-3′	5′-AGCCTTCTCCATGGTGGTGAAGAC-3′
*MMP* *-2*	5′-AGCGAGTGGATGCCGCCTTTAA-3′	5′-CATTCCAGGCATCTGCGATGAG-3′
*MMP* *-9*	5′-GCCACTACTGTGCCTTTGAGTC-3′	5′-CCCTCAGAGAATCGCCAGTACT-3′
*TIMP* *-1*	5′-CTGTTGTTGCTGTGGCTGATAG-3′	5′-CGCTGGTATAAGGTGGTCTGG-3′
*TIMP* *-2*	5′-ACCCTCTGTGACTTCATCGTGC-3′	5′-GGAGATGTAGCACGGGATCATG-3′

## Data Availability

All data are available on request.

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
