# Peer review of "Scorpion Peptide Smp24 Exhibits a Potent Antitumor Effect on Human Lung Cancer Cells by Damaging the Membrane and Cytoskeleton In Vivo and In Vitro"

_toxins, 2022, doi:10.3390/toxins14070438_

Round 1
Reviewer 1 Report
The work “Scorpion peptide Smp24 exhibits a potent antitumor effect on human lung cancer cells by damaging the membrane and cytoskeleton in vivo and in vitro” documents antitumor effects of a scorpion venom peptide in cancer cells and its in vivo effects. All the conclusions made are well supported by the experimental data. However, the following concerns are worth to be commented by authors.
1. I do not see the amino acid sequence of Smp24 either in the main manuscript, or in the Supplementary part. I understand that there are references to previous work on the peptide in the text. However, it would be rather convenient for the reader to find the a.a. sequence,
e.g. in the Figure S1, where RP-HPLC data on the peptide are presented. The lines 311, 312 report: “The high-purity peptide was mixed, lyophilized, and further identified by MALDI-TOF mass spectrometry(Figure S1 and S2).” What means “mixed” in this sentence? May be, we need simply to state that the theoretical mass of the peptide coincides with the experimentally determined one?
2. The choice of the peptide is poorly argumented in the work. In the Discussion section we find references [19-22] to other natural and artificial anticancer peptides. But what is in common between all these peptides? I would suggest to discuss this point in more details, because we find in the end of the article that the peptide Smp24 is patented. But series of peptides are usually patented, not a single one. Probably, in the results of previous experiments some peptides were rejected? More importantly, in the recent literature we find the work: htps://pubmed.ncbi.nlm.nih.gov/35631515/ devoted to Triblock Amphiphilic Short Antimicrobial Peptides. The authors showed that the peptide K4F6K4 affected most strongly A549 cells. But Smp24 has significantly lower overall charge (+3). Thus, the choice of Smp24 to be patented seems strange. Please, discuss these issues in the Discussion section.
3. Continuing to the preceding point, it is not clear what is known about the spatial structure of Smp24 in different environments. What is known from preceding experiments? It is stated in the text that the peptide is amphipathic (e.g. line 51). Does it mean that the peptide form amphipathic helix in the lipid membrane? Please, provide reference(s) to these works, if available. Smp24 affects the cytoskeleton. Does it mean that the peptide can interact directly with the proteins forming the cytoskeleton filaments? For other venom peptides, e.g. spider latarcins it was shown that they interact with multiple targets within the cells due to their flexibility, caused by the absence of disulfide bonds. Smp24 also lacks disulfide bonds. Thus, this seems to be a general property of antimicrobial peptides, which are called linear ones, i.e lacking disulfide bonds? Please, expand discussion to the previous point, paying attention to the structural details of the peptide Smp24.
4. I read (lines 154-156) “As shown in Figure 4E, the expressions of TIMP-1/-2 mRNA were significantly upregulated while the expressions of MMP-2/-9 mRNA were significantly downregulated in a concentration-dependent manner following incubation with Smp24 for 12 h.” I look in Figure 4, E and find (line 187): “Body weight change. “. Is everything correct? Where to find the corresponding data?
5. Some sentences are difficult to understand, e.g. (lines 46-47): “Inhibiting tumor migration is also a strategy of tumor therapy like a short peptide RK1, from Buthus occitanus can inhibits tumor cell migration [6].” Please make the sentence clearer, e.g. by dividing it in two pieces.
Reviewer 2 Report
Reference manuscript TOXINS June 2022: “Scorpion peptide Smp24 exhibits a potent antitumor effect on 2 human lung cancer cells by damaging the membrane and cyto- 3 skeleton in vivo and in vitro”.
General Comments:
In the text presented in this manuscript, the authors are discussing their results that deal with the cytotoxic and antitumor action of the Smp24 toxin, produced by the Egyptian scorpion Maurus palmatus, with emphasis on lung tumor cell lines. Throughout the text, the authors discuss the activities of this toxin on cell viability, disruptive effect on cell membrane, cytoskeleton organization, cell migration, invasiveness, in addition to the regulation of MMP-2, MMP9, TIMP-1 and TIMP-2 expressions. The authors showed that the toxin inhibited growth activity on the A549, H3122, PC-9, and H460 tumor cell lines, but showed little cytotoxicity on the non-malignant MRC-5 lineage. The Smp24 toxin also caused necrosis of A549 cells by destroying the integrity of the cell membrane, mitochondrial and nuclear membrane. In addition, the toxin under study blocked cell migration by altering the cytoskeleton and the expressions of MMP-2, MMP-9 and TIMP-1 and TIMP-2. Finally, the authors showed that the toxin under analysis showed antitumor activity in vivo models using mice, in addition to showing a low activity for animals. The authors conclude by the potential antitumor activity of the toxin and the possibilities of biotechnological applications of its use in antitumor chemotherapy in a future. Undoubtedly, it is a very attractive topic and suitable for publication in TOXINS. The authors dominate the subject and have produced a very clear and relevant text. Nevertheless, before a final acceptance by TOXINS, here are some comments and suggestions that could make a revised text more attractive to readers in the area, thus increasing the impact of the publication. Comments
1- Between lines 65 and 66, the authors wrote .... MTT was used to measure the antitumor activity of Smp24 against four human lung 65 cancer cell lines… In my opinion, in a revised version, the authors could write …to measure the cytotoxic activity, since antitumor activity is a better term to verify in vivo activity, and this was not the case! 2- Generally, and this is elegant and advisable, one must show cytotoxic activities of a given molecule by two methods with different mechanistics. The method used (MTT) is a method that measures metabolism of formazan salts at the intracellular level. Why didn't the authors do a second method, for example that assesses membrane integrity, like Trypan Blue, which is a simple and quick test. 3- About figure 1C, where the authors show morphological changes of A549 cells after treatment with the Smp24 toxin. In my opinion, the magnification of images shown should be larger. As shown, the images do not allow identifying major morphological changes. Also indicate with arrows and other symbols the identified changes. 4- In figure 2B, the images showing cells treated by the Smp24 toxin and analyzed by scanning electron microscopy need to be improved. As shown (a single cell) does not represent statistical confidence. The authors should show more panoramic fields with a greater number of changed cells, proving that the changes are really significant! A single altered cell may be a technical artifact. 5- Once again, figure 2F has the images with very small magnifications! What do the arrows shown in the figures mean? 6- If the authors really want to show that the toxin under study can cause disruption of the nucleus as postulated, why not show images with statistical significance, obtained by Transmission Electron Microscopy, which can provide ultrastructural details of changes in the nuclear membranes, or changes in the chromatin or DNAs.
7- About figure 3A, the images seem out of focus, mainly 3Aa and 3Ab. The magnifications shown also do not allow to see details of alterations in the polymerization of actin filaments as pointed by authors. Finally what are the arrows shown, but not commented on in the caption text?
8- Still about figure 3. what happened to the adhesive potential of these cells with the Extracellular Matrix? If there was disorganization of the actin cytoskeleton, there must also have been changes in focal adhesion points, distribution of integrins, cadherins, among other adhesive molecules. Finally, the morphologies of the treated cells, and in the magnification of images shown, do not seem to indicate loss and disorganization of the actin cytoskeleton, since they were not completely rounded!
9- As for the experiments where the authors show changes in the expressions of Metalloproteases of the Extracellular Matrix MMP 2 and MMP9, it is difficult to comment on the results, as the authors do not show criteria for concentrations of the proteins studied, only mRNAs and this is not enough. They should have performed at least one zymogram with the culture supernatants of the studied cells, and/or immunoWestern blottings with antibodies to the respective MMPs! Since these MMPs are secreted. The Same for expressions of TIMPs, at least one WB. mRNA transcriptions do not mean protein translations!
10- About figure 4, the data are really very interesting, and here we can actually talk about antitumor activity of the tested toxin! However, once again the authors could show the figures of the histopathological analyzes with higher magnifications, where the described alterations were clearer. Also point out these changes with arrows or other symbols, which facilitates interpretation by researchers who are not in the field of histology.
11- About figure 5, I would complete with an assessment of blood count and coagulogram.
12- Na linha 217 completar …..for A549 cell model… 13- At the lines 285 and 286 the authiors wrote …. In conclusion, the present findings indicate that Smp24 exerts an antitumor effect in vitro and in vivo models better to change for … exerts an citotoxic effect in vitro and an antitumor activity in vivo… 14- Finally, I missed the authors discussing the possibility of the tested toxin, which is a protein, being antigenic and generating the production of neutralizing antibodies, which can inactivate the toxin in the bloodstream of exposed animals, after booster doses in the treatments. How to solve this problem to think about future biotechnological applications?
Reviewer 3 Report
The authors explored the in vitro and in vivo potential of the widely studied Smp-24 peptide. In this work, they focus on its antitumor activity against human lung cancer cells.
1. Line 67: What are the standard deviations from the IC50 values? How many independent experiments were performed?
2. Figure 1A. Y axis must end in a number.
3. Quality of figure 1B is low.
4. A reference drug (positive control) should be included in the viability assays for comparison.
5. Lines 66-67: The authors did not describe the results of the peptide's effect on healthy lung cell viability. The Figure 1A shows the 5 cell lines studied, but only 4 IC50s are described, focusing only on cancer lines.
6. The authors should calculate the index of selectively considering the determined IC50.
7. Figure 2F. Please include length bar.
8. The abstract should include some quantitative data, such as concentration, percentages or IC50.
9. Line 277-279. I don't understand why the authors compare the potency or anticancer activity based on peptide length.
10. The number of animals used in antitumor assays was not specified. How was this number of animals defined?
Round 2
Reviewer 1 Report
1. Please, round up all concentrations to numbers with no more than two digits after comma, e.g. 4.061 mkM --> 4.06 mkM.
2. If you want in the Abstract to discuss the tumor cell death mechanism induced by Smp24, you need to take into account that in KG1-a cells the mechanism is pyroptotic, not membrane-disrupting ( https://pubmed.ncbi.nlm.nih.gov/35082671/). Otherwise, delete the words "by membrane disruption" from the first sentence of the Abstract (line 7).
3. The word "mass" is missing in the sentence, lines 360-361. Probably, you mean: "The theoretical mass of 2578.09 Da of the peptide is close to the experimentally determined one, 2578.3 Da."
4. English editing is still required, e.g. line 324: "Smp24 is low cost and efficient than melittin". You mean: " is more efficient than melittin, being cheaper"?
Author Response
1.Thanks for your meaningful suggestion. We have revised all data as you suggested.
2.Thanks for your meaningful suggestion. It is really important for our manuscript. We neglected the paper published in this year and forgot to update our manuscript which were well prepared last year. Now we have revised it in our new revision. Thanks again!
3.We greatly thank for your kindly mention and we have corrected the written carelessness in the revised manuscript.
4.We greatly thank for your kindly mention and we have corrected the written carelessness in the revised manuscript. The paragraph in line 324 focuses on the advantages of Smp24 in comparison with melittin. Smp24 shows stronger in-vivo antitumor capacity while shorter sequence than melittin. Thus, Smp24 is low synthesis cost and efficient than melittin. Since this expression was misleading, we have carefully rewritten the sentence to avoid this type of misunderstanding in revision manuscript.

Reviewer 2 Report
After careful reading of the letter written by the authors, where they responded in an elegant, professional and convincing manner to all considerations raised by the reviewer, it is my opinion that this revised form of the manuscript may be approved for publication in Toxins.
Reviewer 3 Report
The revised version of the manuscript is improved. Therefore, I am now supportive of its publication.
Author Response
We appreciate for your warm work earnestly and comments.